



# Inter- and Intra-annual Surface Velocity Variations at the Southern Grounding Line of Amery Ice Shelf from 2014 to 2018

Zhaohui Chi[1], Andrew G. Klein[1]

[1]Department of Geography, Texas A&M University, College Station, Texas, 77845-3147, USA

*Correspondence to*: Zhaohui Chi (zchi@tamu.edu)

**Abstract.** The ice flow rate through the grounding line of the Amery Ice Shelf (AIS) is vital to understanding the mass discharge received from its three primary tributary glaciers. Previous studies have indicated a stable multiyear average surface velocity distribution in the convergence area of AIS. However, the surface velocity variations, especially short-term fluctuations, in the AIS have been relatively undocumented. This study investigated inter-annual and intra-annual surface velocity variations along the southern segment of AIS grounding line from 2014 to 2018. Using feature tracking to derive surface velocity for five consecutive austral summers and winters, it was found that AIS's upstream end has experienced a steady ~ 5% inter-annual increase in surface velocity. Surface velocity increases were observed in 2014/2015 (0.25±0.02 m/d) and in 2017/2018 (0.21±0.02 m/d) respectively. Surface velocities in winters were lower than the summers except for 2016, which had a 0.12 m/d surface velocity decrease from winter to summer. Although flowing slower than the other two glaciers, Fisher Glacier exhibited the highest inter-annual increase (8.56±4.36%) and the largest intra-annual variation (-5.41±5.65%) in surface velocity of the three studied glaciers. While the surface velocity observed in 2018 was generally close to the observed velocity in 1989, the magnitude of velocity variations observed during the 2014–2018 period is similar to the differences in velocities measured at the grounding line since 1989. This indicates continued relative stability in the surface velocities at the grounding line of these three tributary glaciers but also indicates that caution should be applied when interpreting long-term differences based on a limited number of measurements. This study demonstrated the capability of feature tracking to monitor the multidecadal changes of surface velocity.

## 1 Introduction

Surface velocity measurements are important for understanding the motion of glaciers and ice sheets in Antarctica. It is essential to determining ice redistribution from the interior regions toward the ocean as well as their contributions to sea level rise (Rignot et al., 2008; Rignot et al., 2011b). The Intergovernmental Panel on Climate Change Report (IPCC, 2013) suggested a probable response of ice shelves to a warming climate is an enhanced surface and basal melting, thereby driven by which increasing ice surface velocity. In part due to enhanced ice velocity, increased ice discharge has been considered as a contributor to globally observed mass losses (Rignot et al., 2004; Rignot and Kanagaratnam, 2006). In response to global climate changes particularly in recent years, ice flow in Antarctica has been found to be increasingly driven by accelerated surface and basal melting along with the loss of buttressing ice shelf front (Pritchard et al., 2012; Miles et al., 2013; Shen et al., 2018). New computational models for estimating the ice sheet mass budget have been published, but many of them require accurate ice flow change measurements (Gillet-





Chaulet et al., 2012; Seddik et al., 2012; Enderlin et al., 2014). Thus long-term accurate velocity measurements remain
fundamental to refining uncertainties in the estimates of mass balance of the Antarctic Ice Sheet.

As the largest ice shelf in East Antarctic Ice Sheet, the Amery Ice Shelf (AIS) buttresses the Lambert Glacier Basin
which constitutes the fourth largest drainage system in Antarctica (Giovinetto, 1964; McIntyre, 1985) draining
approximately 12.5% of the Antarctic Ice Sheet (Drewry, 1983). Lambert Glacier and other two primary tributary
glaciers transport ice from the interior Lambert Glacier Basin through the southern grounding line of the AIS
discharging into the Prydz Bay. This large drainage area occupies only 2.5% of the East Antarctic coastline (Fricker
et al., 2000) making AIS a sensitive indicator to mass balance change of the East Antarctic Ice Sheet. Furthermore,
due to the fast ice flows are primarily located along the southern segments of the AIS grounding line this area becomes
an important ice fluxgate to monitor the majority of ice mass redistribution into the AIS.

Remote sensing has revolutionized our ability to observe and measure surface velocities over vast areas with high
spatial resolution and has provided the comprehensive observations needed for modern scientific investigations of ice
dynamics and mass balance. Many early surface velocity estimates were made by expensive field survey
measurements (Stephenson and Bindschadler, 1994; Frezzotti et al., 1998). Field survey techniques, particularly with
the usage of Global Positioning System (GPS), have been used later (Manson et al., 2000; Sunil et al., 2007; Zhang et
al., 2008), but remain logistical difficulty in accessing remote portions of Antarctica and make them expensive to
obtain (Yang et al., 2014). Recent decades have seen the benefit from the advent of interferometric SAR (InSAR)
which enables accurate surface motion detection over large areas (Goldstein et al., 1993; Rosen et al., 2000; Joughin,
2002; Young & Hyland, 2002; Yu, 2005; Tang, 2007). However, successful InSAR measurements remain highly
dependent on the data acquisition, topographic characteristics, and coherence of SAR image pairs.

Feature tracking is a widely used image-based technique which tracks the motion of distinctive surface features
moving with the ice and persistent in image pairs acquired by space-borne optical instruments over time (Gray et al.,
1998). Successfully tracking the motion of small surface features through cross-correlation (Bindschadler et al., 1994;
Berthier et al., 2003; Stearns and Hamilton, 2005; Fahnestock et al., 2016) requires that the same feature be detected
reliably and consistently across sequential images. The extremely frozen environment in Antarctica enables to retain
consistent surface features well on the ground surface, which is suitable for a feature tracking approach.

Comprising the largest glacier-ice shelf system in East Antarctica, three tributary glaciers along with the AIS have
been investigated by surface velocity mapping (Joughin, 2002; Yu et al., 2010; Pittard et al., 2012; Pittard et al., 2015).
However, the surface velocity variations in the AIS have been relatively undocumented, with limited research
suggesting a stable surface velocity distribution and no loss of ice mass exhibit in the AIS (Pittard et al., 2015; Pittard
et al., 2017). This study uses feature tracking to investigate the inter-annual and intra-annual surface velocity variations
along the southern grounding line of AIS. Here surface velocities are measured for five consecutive austral summers
and winters between 2014 and 2018.



## 2 Study Area

The AIS (Fig 1) is geographically located in East Antarctica spanning approximately 68.5°S to 81°S latitude and 40°E to 95°E longitude (Fricker et al., 2000). The Lambert, Mellor, and Fisher Glaciers originate farthest inland, flow northward, and form a confluence zone discharging to AIS through the fluxgate between the Prince Charles Mountains and the Mawson Escarpment (Hambrey, 1991; Yu, 2005). The southern grounding line is the location of the ice fluxgate connecting the interior Lambert Glacier Basin with a large embayment northward of Prydz Bay.

## 3 Data Preparation

The Landsat-8 satellite was successfully launched by the National Aeronautics and Space Administration (NASA) in 2013 with the orbit height of 705 km, the orbital inclination of 98.2°, the orbital period of 98.2 minutes, and an 8-day revisit cycle. Its Operational Land Imager (OLI) image product consists of 7 spectral bands at 30 m and a panchromatic band (band 8) at 15 m spatial resolution respectively. Provided by the United States Geological Survey (USGS), the Landsat-8 L1GT product has been co-registered with the high-resolution Radarsat Antarctic Mapping Project (RAMP) Version 2 DEM to enable systematic geometric terrain correction (USGS, 2019) and was used in this study. The L1GT product is stored as 16-bit signed integers.

Ten image pairs constructed from twenty cloud free images (Table 1) were used to derive the surface velocities for the consecutive austral summers and winters during 2014–2018. An additional cloud free image pair was used to derive the latest surface velocity for 2019 along with the determination of the locations of the ends of the southern grounding line segments of the AIS used in this study. Winter is defined as July–October and summer as January–March. The preferred temporal interval for each image pair is one month but the actual time separation varies slightly due to the data availability. All images were processed using Environment for Visualizing Images (ENVI5.5) software package supplemented an installation of the Co-registration of Optically Sensed Images and Correlation (Cosi-Corr) program (Leprince et al., 2007).

In this study, the grounding line was determined by the Mosaic of Antarctica (MOA) 2008–2009 mission (Haran et al., 2014). This MOA grounding line product was generated using the composited Moderate Resolution Imaging Spectroradiometer (MODIS) image data acquired during the 2009 austral summer. The ice velocity and Antarctic basin boundary were provided by the Making Earth System Data Records for Use in Research Environments (MEaSUREs) program. The MEaSUREs surface velocities Version 2 (Rignot et al., 2017) observed in the study area were an InSAR-based multiyear average velocity during 1996–2016.

## 4 Methods

As illustrated in Fig 2, feature tracking is based on an image-to-image cross-correlation algorithm which locates the identical surface features from sequential images and measures their displacements in the frequency domain (Bindschadler & Scambos, 1991; Scambos et al., 1992). This algorithm assumes: (1) a pre-event image (reference



image) plus a shift translation comprising a post-event image (search image); (2) the reference image and its paired
search image are assumed to share the same ground resolution and (3) are geolocated.

Surface features are represented by the patterns of a group of individual pixels. By shifting a 64 x 64 pixel search
window across the images in a pair (Ayoub et al., 2009b) every 4 pixels, the displacement of the dominant feature
within the window is computed through the normalized covariance correlation method (Bernstein, 1983). In the same
corresponding central pixel, the north-south (N/S) and east-west (E/W) components of the correlation are recorded, as
well as the signal-to-noise ratio (SNR), which is taken as the ratio of peak correlation function to the average value.

Given by two motion estimate components, N/S and E/W, the surface velocity can be calculated using Eq. (1):

$$v = SQRT(NS^2 + EW^2)/t \tag{1}$$

where $v$ represents the surface velocity magnitude estimate, $SQRT(NS^2 + EW^2)$ is the displacement magnitude, and $t$
represents the time interval between acquisition dates.

Due to providing sharp surface texture at 15 m resolution the Landsat-8 OLI band 8 imagery was used for feature
tracking. Preprocessing included applying a 3x3 Gaussian high pass filter to enhance the distinct surface features from
the panchromatic band 8 for both reference and search images. The Cosi-Corr software package was implemented to
accomplish the feature tracking procedures. This software package is a robust feature tracking program to obtain
surface velocity measurements and is available from the Caltech Tectonics Observatory website
(http://www.tectonics.caltech.edu/). Full details of the package can be found in Leprince et al (2007).

A set of location reference points were generated at 0.1 km intervals along the ~76 km AIS grounding line to facilitate
the monitoring of surface velocity variations as well as the accuracy and uncertainty assessments. Surface velocity
measurements were extracted using the geographic coordinates of the location reference points. The accuracy of the
surface velocity measurements was assessed using signal-to-noise (SNR) ratio as SNR value refers to the correlation
between the image window and the search window (Ayoub et al., 2009b). The SNR value of 1 represents the two
windows are perfectly correlated, while the SNR value of 0 shows no correlation.

All the computed velocities were passed through a quality control procedure to reduce possible miscorrelated
estimations. Any magnitude variation greater than 5% was considered to be an outlier and was disregarded in further
analysis. The magnitude variation was defined as the difference between the measurement and the mean of the two
adjacent measurements at the grounding line. Observation with low SNR (< 0.9) was omitted as well due to possible
miscorrelation.

Uncertainty analysis is fundamental to estimate the applicability of the determined surface velocity measurements.
Uncertainty of a single-pair velocity measurement is dominated by the accuracy of the feature tracking algorithm, the
co-registration of pre- and post- images, and the temporal variability of ice surface (Dehecq et al., 2015). The L1GT





Landsat-8 OLI product used by this study has been applied with precision terrain correction and the uncertainty of
imagery co-registration has been minimized The uncertainty coming from the temporal variability of ice surface can
be negligible, since only the images with identical surface features in both pre- and post- images are used. Hence, the
uncertainty of the feature tracking algorithm exhibits in all the final velocity measurements and needs to be assessed
quantitatively.

The velocity measurement uncertainty can be assessed through the observed shifts in stable zones (e.g. exposed rocks,
massif, etc). However, due to inconsistent acquisition angles and temporal changes, any difference (e.g. shadow and
illumination as illustrated in Fig 3) exhibiting between pre- and post-event images can cause a measurable
miscorrelation. Therefore a small stable ice surface located in the north end of a nunatak was used in this study for
uncertainty assessment as no motion in this stable area should occur. Fifty random locations were generated within
the boundary of the selected stable ice surface (indicated in red dot in Fig 3).
**5. Results**
**5.1 Major Ice Stream Velocity Identification**
From the MOA grounding line product, the two ends of the AIS's southern grounding line segment for this study were
established where the surface velocity decreased to 0.5 m/d, based on the surface velocity measurements for 2019 (Fig
4). The grounding line was divided into three segments according to the MEaSUREs Antarctic basin boundaries
(Rignot et al., 2013) as illustrated in Fig 4A. The lengths of each segment are 35.6 km, 14.1 km, and 26.3 km for
Fisher Glacier, Mellor Glacier, and Lambert Glacier, respectively. The mean velocity along a 3 km long grounding
line centered on the peak velocity was then computed to represent the highest surface velocity observed for each
glacier. The analysis for each individual glacier is discussed in section 5.4 is based on their average peak velocity as
illustrated in Fig 4B.
**5.2 Inter-annual Surface Velocity Variations**
Using feature tracking applied to Landsat-8 OLI band 8 image pairs, ice surface summer and winter velocities were
measured annually from 2014–2018. Each velocity measurement was generated using a pair of preprocessed pre- and
post-event images and represents an average surface velocity during the image acquisition interval.

The annual surface velocity was determined as the mean of the corresponding summer and winter velocities for each
year. The inter-annual variation was calculated by the surface velocity difference of the latter year and prior year over
the prior year velocity. A positive inter-annual variation indicates a surface velocity increase and vice versa.

Overall, the surface velocities across all three glaciers show a consistent ~ 5% inter-annual increase over the study
period (Fig 5). The multiyear mean surface velocities of Fisher Glacier, Mellor Glacier, and Lambert Glacier were
1.31, 1.93, and 2.17 m/d respectively during 2014–2018. The inter-annual surface velocity increased on average
0.09±0.02 m/d (equivalent to 6.5%) over the study period (Table 2). All the three glaciers exhibited on average inter-





annual surface velocity increase, which was 0.08±0.02 m/d, 0.10±0.02 m/d, and 0.10±0.03 m/d for Fisher, Mellor, and
Lambert Glaciers, respectively. Most of the inter-annual surface velocity increase of Mellor Glacier was from a large
inter-annual increase of ~ 0.31 m/d during 2014/2015.
**5.3 Intra-annual Surface Velocity Variations**
Two additional parameters, intra-annual variation and inter-seasonal variation, were calculated for each individual
glacier describing its surface velocity variations. Intra-annual variation is defined as the velocity variation calculated
by the summer-winter surface velocity difference over the summer surface velocity. A negative intra-annual variation
indicates the winter velocity is less than the summer velocity and vice versa. Inter-seasonal variation is defined as the
velocity variation calculated in percentage by the surface velocity difference between the current season (winter or
summer) and the prior season (summer or winter) over the current season's velocity. A positive inter-seasonal variation
represents the current season velocity is higher than the prior season velocity and vice versa.

The multiyear mean measurements indicate that on average surface velocities for winters and summers are similar
(Fig 6A). In addition to the changes in magnitude, the spatial pattern of velocity varied over the study period as well.
While the spatial pattern of surface velocities across the grounding lines of Fisher and Mellor Glaciers were generally
uniform both inter- and intra-annually, two regions of increased velocity appeared along the Lambert Glacier's
grounding line during the summer of 2018 (Fig 6B).

Except for 2016 and 2018, consistent variations in surface velocity were observed for Fisher Glacier and Mellor
Glacier with the winter surface velocity being higher than the summer velocity (Fig 6C). The winter surface velocity
of Lambert Glacier was faster than the summer surface velocity for two consecutive years and then declined in 2018.
While the surface velocities of Fisher Glacier and Mellor Glacier were higher in winter than in summer of 2018, the
opposite was true for Lambert Glacier. A significant inter-seasonal surface velocity increase was observed between
2014 winter and 2015 summer for all the three studied glaciers, while the largest increase of 64.3% was observed in
Fisher Glacier. An increase in inter-seasonal surface velocity was observed again in the summer of 2018 at Lambert
Glacier.

On average the intra-annual surface velocity variation was -0.06±0.10 m/d over the study period, indicating slightly
lower average surface velocity in winter of approximately 3.13% (Table 3). The average intra-annual surface velocity
variation of Fisher Glacier, Mellor Glacier, and Lambert Glacier was small at -0.08±0.06 m/d, -0.02±0.09 m/d, and -
0.07±0.14 m/d respectively. This is consistent with the overall increase in velocities over the study period. Most of
the intra-annual surface velocity variation was contributed by Fisher Glacier due to its large intra-annual surface
velocity variation of -0.42±0.04 m/d in 2015. A positive intra-annual surface velocity variation was observed in 2016
and 2018 for all the three studied glaciers, which indicates the glaciers may even flow faster in winter than in summer
for these two years.
**5.4 Individual Glacier Analysis**



Since the surface velocity of each glacier varies considerably along the grounding line, the surface velocities for the
fastest moving segments are underrepresented by the average surface velocity over the entire grounding line.
Therefore, the average surface velocity at the peak location (illustrated in Fig 4) was computed and used to investigate
the inter-annual variation for each individual glacier. The inter-annual variation was converted into percentage by
dividing the prior surface velocity measurement. Annual surface velocities were generally stable across the three
studied glaciers (Fig 7A), but the inter-annual variation revealed a clear surface velocity decline during 2015/2016
(Fig 7B).
Although flowing slower than the other two glaciers, the surface velocity of Fisher Glacier was found to have greater
intra-annual and inter-annual variability. The intra-annual surface velocity variation on average was -5.41±5.65%,
which indicates lower winter than summer surface velocities (Table 4). However, the average intra-annual surface
velocity variation of Fisher Glacier was 6.86%, indicating that on average the annual surface velocity of Fisher Glacier
increased 8.56±4.36% every year. The largest intra-annual surface velocity variation of Fisher Glacier was -
30.32±4.51 % (-0.25 m/d) in 2015.
Mellor Glacier had intermediate flow velocities between the other two glaciers and the inter-annual surface velocity
increased 6.47±3.48% (0.41 m/d) over the study period, which is the medium variation among three studied glaciers
and can be translated into a 150 m per year velocity increase. Also, compared with the other two glaciers, Mellor
Glacier had the smallest inter-annual surface velocity increase in the last few years, which was 2.96±3.48% during
2016–2017 and 10.15±1.65% during 2017–2018.
As the fastest flowing tributary glacier, Lambert Glacier possessed the largest multiyear average inter-annual surface
velocity, which was 2.17 m/d. There was only -3.28±9.47% intra-annual variation and 5.79±9.20% inter-annual
increase of surface velocity observed in Lambert Glacier, which were the lowest of the three. However, the fastest
inter-annual increase of surface velocity with high uncertainty was found in Lambert Glacier in the last years of the
observation period, which was 7.73±7.75% during 2016/2017 and 17.24±15.06% during 2017/2018.
**5.5 Accuracy and Uncertainty Assessments**
The overall accuracy of feature tracking is approximately 96.5%, according to the accuracy analysis shown in Table
5. Besides a consistent high accuracy exhibiting over the studied time period, there is no significant differences in the
measured uncertainty between summers and winters.
A small stable ice surface adjacent to an exposed rock island (indicated in Fig 3) was selected for uncertainty
assessment due to its minimal illumination differences between images. Uncertainty assessment of the feature tracking
algorithm was conducted based on fifty random locations. The 'box and whisker plot' method was used to visualize
the uncertainty for each year and the average uncertainty along the N/S and E/W directions. A generally close
uncertainty was observed in each individual study year (Fig 8A). The overall uncertainty was 0.20 m/d, while the





overall uncertainty along the N/S and E/W directions are 0.00±0.02 m/d and -0.04±0.01 m/d respectively (positive
towards East and negative towards West; Fig 8B). As shown in Fig 8A, the largest uncertainty at 0.25±0.01 m/d was
observed in 2014, while the smallest uncertainty at 0.15±0.09 m/d exhibited in 2018. We investigated the contribution
of the measured magnitudes and directions of these uncertainties in our computed velocity and variability assessments
and found that they did not lead to any significant adjustments to our measurements.
**6 Discussion**
There is currently little relevant research documenting inter-annual and seasonal changes in the surface velocity in the
AIS. However, there are a number of studies documenting such inter-annual and seasonal variations elsewhere in East
Antarctica. A 9.9 m/yr (equivalent to 0.02 m/d) inter-annual increase in surface velocity was observed in Langhovde
Glacier, East Antarctica from 2008–2010 (Fukuda et al., 2014). Zhou et al. (2014) reported a 19% seasonal decrease
in surface velocity from summer to winter during 1996–2008 in Polar Record Glacier (PRG), East Antarctica, which
is a small outlet glacier on the east side of the AIS towards the southern shore of Prydz Bay (Liang et al., 2019). This
outlet glacier experienced velocity variations of up to 15% from 2005–2015 (Liang et al., 2019), which is equivalent
to a 1.5% inter-annual increase. In addition, Liang et al. (2019) also demonstrated up to 9% seasonal increase in
surface velocity between winter and summer in PRG, East Antarctica.

This research demonstrates that the primary tributary glaciers into the AIS exhibit inter-annual, intra-annual, and inter-
seasonal variations in surface velocity approaching 6.94±5.68%, -2.98±6.67%, and 3.67±9.82%, respectively, along
the grounding line over the five year period 2014–2018 which is consistent with previous studies. The uncertainty
remains high given the limited five-year comparison window. However, over this period there is an overall 0.09±0.02
m/d increase in inter-annual surface velocity variation translating into a ~29 m per year velocity increase or
approximately a 5% increase in average velocity. The overall intra-annual surface velocity variation at -0.06±0.02 m/d
shows that on average winter surface velocity is ~15 m per year slower than summer surface velocity. No obvious
velocity variations appear to exist between summers and winters from our multiyear average measurements. Overall,
a consistent inter-seasonal surface velocity variation has been observed across all the three glaciers excluding a 9.1%
decrease in velocity of Lambert Glacier from summer to winter in 2018. A significant surface velocity increase was
observed across all the three glaciers between 2014 winter and 2015 summer, including the largest 64.3% increase in
surface velocity in Fisher Glacier.

It appears that the areas along the grounding line experiencing the highest velocities are also most sensitive to the
inter-annual and intra-annual velocity variations than the average velocity across the entire ground line of the three
studied tributary glaciers. Fast moving ice flow is more sensitive to the local environmental changes compared to slow
moving ice flow as glacier flow speeds are known to be an important factor in governing glacier's response to the
local environmental changes (Scambos et al., 2004; Stearns et al., 2008; Davies et al., 2012). Fisher Glacier
experiences greater velocity variations than the other two tributary glaciers most probably due to its slower velocities
and smaller size.




Our measured velocities over the 2014–2018 period are in general agreement with previous published results (Rignot
et al., 2017) for earlier decades. While limited historic surface velocity variations from either InSAR or feature-
tracking, those that do exist are in general agreement with the surface velocity measurements we observed over our
five-year study. As illustrated in Fig 9A, a good agreement exhibited in both the magnitude and location of the peak
flow velocity for all the three glaciers.

Based on the limited observations, a surface velocity decline appeared from 1989 to the early 2000s which continued
through 2014. From 2014–2018 the surface velocities were found to increase. In 2018 the surface velocity was
generally close to the velocity observed in 1989 (Fig 9B) and the locations in peak flow velocities along the grounding
line particularly for Fisher Glacier and Lambert Glacier were largely unchanged. However, it is important to note that
the variations in velocity observed in this study between 2014 and 2018 encompass nearly as large a range in those
decadal differences observed from 1989 to present.
**7 Conclusions**
This study examines surface velocity variations of temporal and spatial distribution along the southern segment of the
AIS grounding line. Furthermore, feature tracking approach is suitable to determine the surface velocity in the study
area, achieving the accuracy comparable to the InSAR method.

The multiyear mean feature tracking-based surface velocity is 1.31, 1.93, and 2.17 m/d for Fisher Glacier, Mellor
Glacier, and Lambert Glacier respectively. This study reveals a ~ 5% (0.09±0.02 m/d) annual increase during 2014–
2018. Lambert Glacier and Mellor Glacier exhibit higher average inter-annual surface velocity increase at ~0.10 m/d,
while Fisher Glacier showed the largest on average intra-annual surface velocity at -0.08±0.06 m/d. The studied
glaciers flow in winter nearly the same fast as in summer with only 0.07±0.14 m/d difference within each year which
is only 3% of their average annual velocity.

These observations indicate variability in velocities along these grounding lines, but over the five year study period
general stability in surface velocities were observed. As the variations in surface velocities observed for these three
major tributary glaciers over the study period of 2014–2018 are similar in magnitude to the range of velocities to that
of measured since 1989 of the AIS suggests caution needs to be applied when comparing limited measurements from
various decades. This also suggests the surface velocities along the grounding line of the major tributary glaciers of
the AIS have remained relatively stable since the late 1980s.
**Acknowledgements**
We are thankful for the National Snow & Ice Data Center (NSIDC) for publishing the grounding line data at
https://nsidc.org/data/nsidc-0593. We also would like to thank the USGS for making the Landsat archive freely
available at http://earthexplorer.usgs.gov/.



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

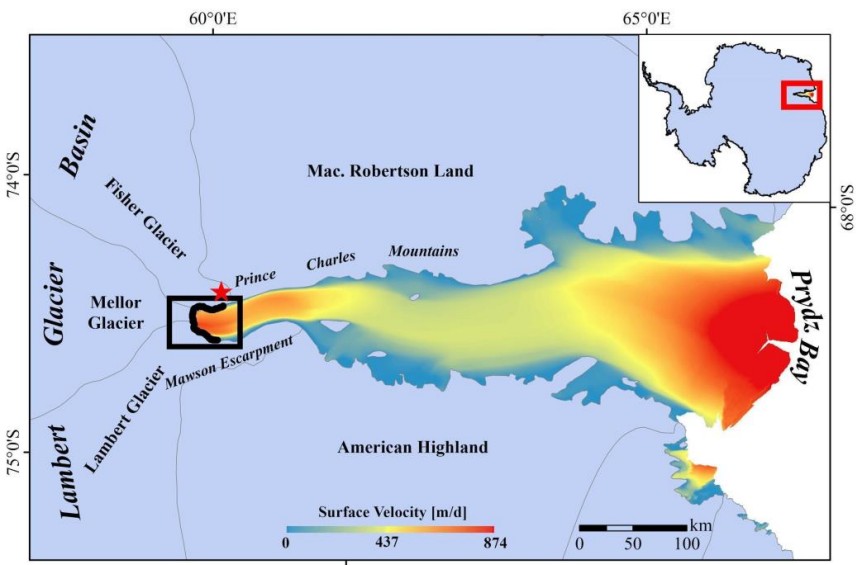


**Figure 1: Location of southern segmentation of the AIS grounding line (~76 km in length) highlighted in bold black line.**
**The background image is the InSAR-derived ice velocity map (Rignot et al., 2017) masked by the basin boundaries. The**
**black box represents the location of the study area. The red star is the location selected for uncertainty assessment (Fig 3).**
**The red box in the inset map shows the location of Fig 1.**




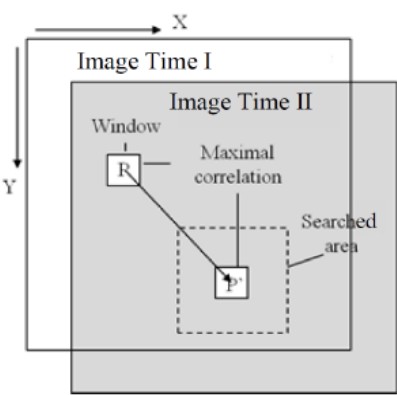


**Figure 2: Sketch map illustrating feature tracking (Chi, 2012).**

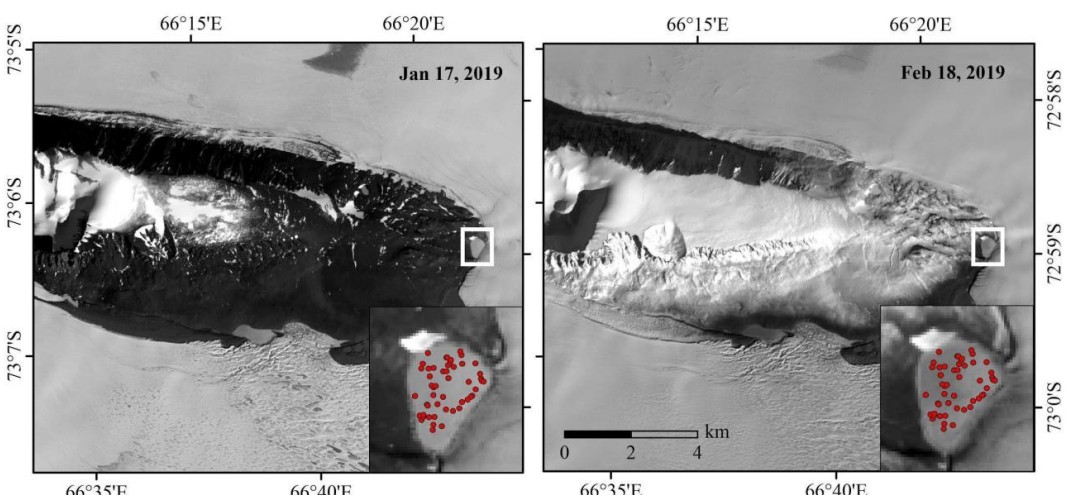


**Figure 3: The location used for uncertainty assessment is highlighted in white box superimposed on the Landsat-8 OLI**
**band 8 image acquired on January 17, 2019 (Left) and on February 18, 2019 (Right). The inset maps illustrate the fifty**
**random locations generated for uncertainty assessment. The location of Fig 3 has also been shown in Fig 1.**


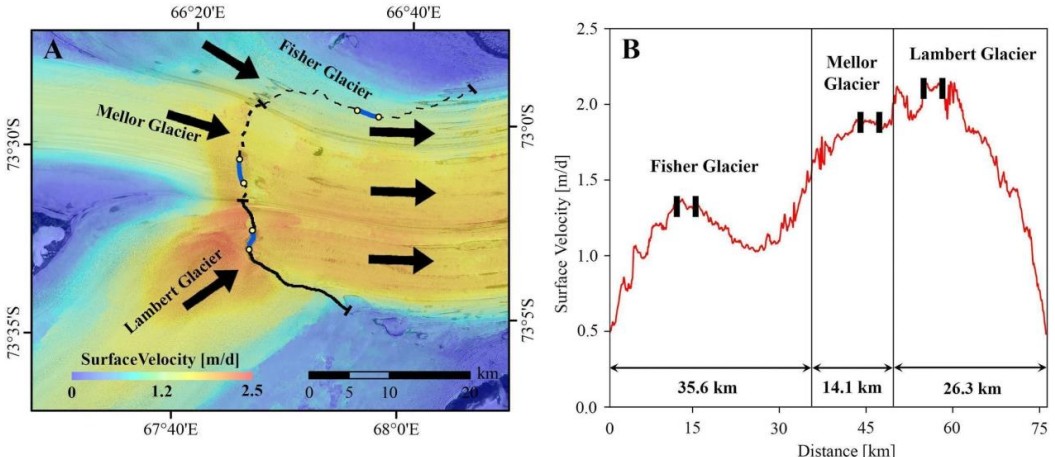


**Figure 4: (A) Grounding line segments corresponding to the Fisher Glacier, Mellor Glacier, and Lambert Glacier are illustrated using different line symbols superimposed on the feature tracking-based surface velocity map using Landsat-8 OLI image pair of 01/17/2019 and 02/18/2019. Black arrows represent the ice flow direction. (B) The corresponding surface velocity profile along the studied grounding line segmentation. The grounding line segments highlighted in blue (Fig 4A) correspond to the black parallel lines representing the peak velocity locations.**

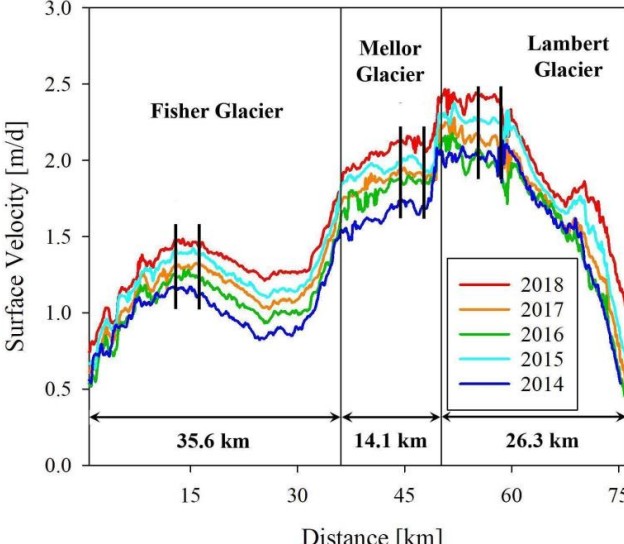

528

**Figure 5: Surface-parallel annual surface velocity during 2014–2018. Black parallel lines represent the peak velocity location for each studied glacier.**

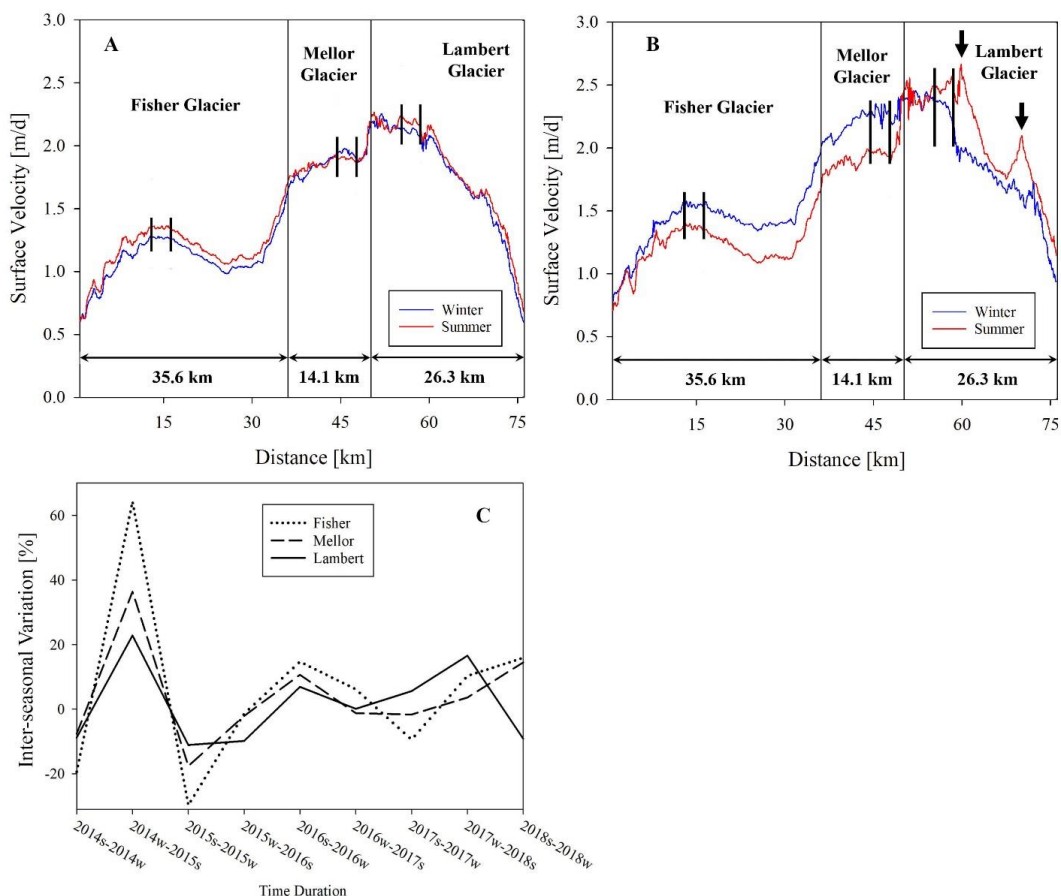

**Figure 6: Surface velocity profiles of (A) multiyear average for the summers and winters, (B) the year of 2018, and (C) the inter-seasonal variations for five consecutive summers and winters during 2014–2018. Note the x-axis label in Fig 6c represents prior-current season. Black arrow indicates the new region of increased velocity identified by this study.**

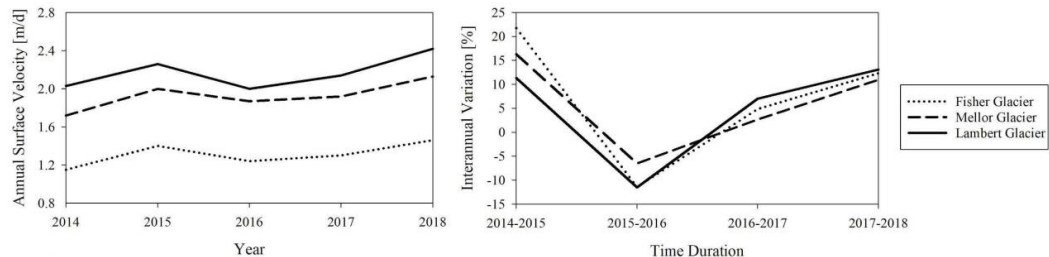

**Figure 7. (A) Annual surface velocity and (B) inter-annual surface velocity variation at the peak velocity location for the three studied glaciers.**



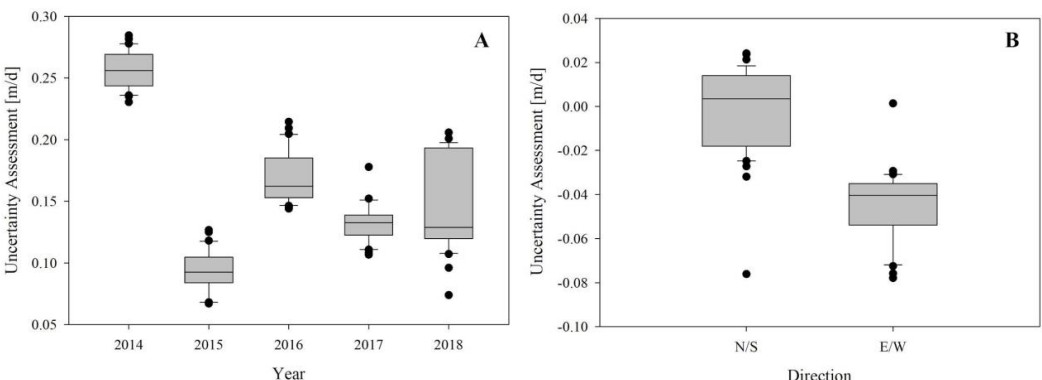

**Figure 8: The uncertainty assessment represented by the box-and-whisker chart (A) for each year and (B) along N/S and E/W directions respectively during 2014–2018.**

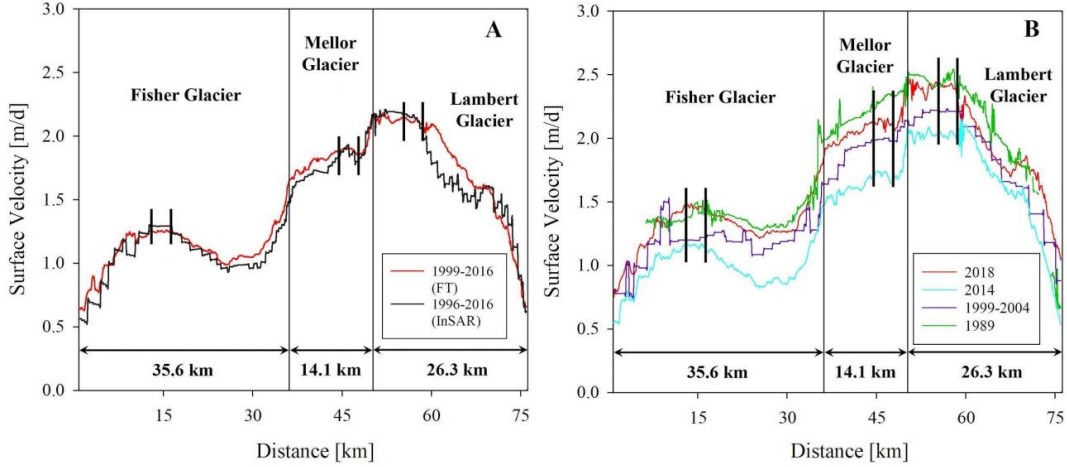

**Figure 9: Comparisons of surface velocity (A) of the multiyear mean estimates between the feature tracking velocity and the MEaSURE InSAR velocity at version 2 (Rignot et al., 2017), and (B) of the multi-temporal annual estimates derived using feature tracking alone. Note that the historical surface velocities (1989 & 1999–2004) were provided by Chi's dissertation work (2012). FT stands for feature tracking.**



**Table 1: List of Landsat-8 OLI image pairs acquired for feature tracking.**

| Pair | Pre-Date | Post-Date | Time Interval (d) |
|------|----------|-----------|-------------------|
| 1 | 01/17/2019 | 02/18/2019 | 32 |
| 2 | 09/27/2018 | 10/22/2018 | 25 |
| 3 | 01/23/2018 | 02/22/2018 | 30 |
| 4 | 09/17/2017 | 10/26/2017 | 39 |
| 5 | 01/27/2017 | 02/28/2017 | 32 |
| 6 | 09/21/2016 | 10/23/2016 | 32 |
| 7 | 02/17/2016 | 03/13/2016 | 25 |
| 8 | 10/05/2015 | 11/13/2015 | 39 |
| 9 | 01/22/2015 | 03/02/2015 | 39 |
| 10 | 10/09/2014 | 11/19/2014 | 41 |
| 11 | 02/02/2014 | 02/27/2014 | 25 |



**Table 2: Inter-annual variations of peak surface velocity for each studied glacier (m/d).**

| Year | Fisher | Mellor | Lambert | Mean±Stdev |
|------|--------|--------|---------|------------|
| 2014/2015 | 0.24±0.02 | 0.29±0.02 | 0.23±0.01 | 0.25±0.02 |
| 2015/2016 | -0.15±0.02 | -0.13±0.02 | -0.26±0.04 | -0.18±0.03 |
| 2016/2017 | 0.06±0.02 | 0.05±0.02 | 0.13±0.04 | 0.08±0.03 |
| 2017/2018 | 0.16±0.01 | 0.20±0.03 | 0.28±0.02 | 0.21±0.02 |
| Mean±Stdev. | 0.08±0.02 | 0.10±0.02 | 0.10±0.03 | **0.09±0.02** |


**Table 3: Intra-annual variation of surface velocity for each studied glacier (m/d).**

| Year | Fisher | Mellor | Lambert | Mean±Stdev |
|------|--------|--------|---------|------------|
| 2014 | -0.21±0.06 | -0.13±0.09 | -0.16±0.21 | -0.17±0.12 |
| 2015 | -0.42±0.04 | -0.38±0.09 | -0.22±0.09 | -0.34±0.07 |
| 2016 | 0.15±0.05 | 0.18±0.12 | 0.11±0.15 | 0.15±0.11 |
| 2017 | -0.11±0.07 | -0.03±0.07 | 0.10±0.09 | -0.01±0.08 |
| 2018 | 0.19±0.08 | 0.28±0.08 | -0.19±0.17 | 0.09±0.11 |
| Mean±Stdev. | -0.08±0.06 | -0.02±0.09 | -0.07±0.14 | **-0.06±0.10** |






**Table 4: Intra-annual and inter-annual surface velocity variations for three studied glaciers.**

| Year | Intra-annual Variation (%) | | | | Inter-annual Variation (%) | | | |
|---|---|---|---|---|---|---|---|---|
| | Fisher | Mellor | Lambert | Mean | Fisher | Mellor | Lambert | Mean±Stdev |
| 2014 | -19.13 ±5.58 | -7.66 ±5.62 | -10.62 ±16.79 | -12.47 ±9.33 | | | | |
| 2015 | -30.32 ±4.51 | -17.56 ±4.48 | -11.71 ±5.66 | -19.86 ±4.88 | 25.02 ±5.13 | 19.63 ±4.70 | 11.52 ±6.05 | 18.72 ±5.29 |
| 2016 | 15.79 ±7.38 | 11.07 ±6.25 | 8.96 ±10.31 | 11.94 ±7.98 | -13.06 ±3.39 | -6.86 ±4.11 | -13.33 ±7.95 | -11.08 ±5.15 |
| 2017 | -9.17 ±4.25 | -1.73 ±3.84 | 6.33 ±7.24 | -1.52 ±5.11 | 8.44 ±4.24 | 2.96 ±3.48 | 7.73 ±7.75 | 6.38 ±5.15 |
| 2018 | 15.77 ±6.54 | 14.66 ±4.34 | -9.34 ±7.33 | 7.03 ±6.07 | 13.83 ±4.69 | 10.15 ±1.65 | 17.24 ±15.06 | 13.74 ±7.13 |
| Mean ±Stdev | -5.41 ±5.65 | -0.24 ±4.90 | -3.28 ±9.47 | **-2.98** **±6.67** | 8.56 ±4.36 | 6.47 ±3.48 | 5.79 ±9.20 | **6.94** **±5.68** |


**Table 5: SNR of surface velocity measurements along the southern grounding line of AIS during 2014–2018.**

**OA stands for overall accuracy.**

| Year | Summer | Winter | Annual |
|---|---|---|---|
| 2014 | 0.9692 | 0.9730 | 0.9711 |
| 2015 | 0.9601 | 0.9658 | 0.9630 |
| 2016 | 0.9744 | 0.9603 | 0.9674 |
| 2017 | 0.9630 | 0.9612 | 0.9621 |
| 2018 | 0.9542 | 0.9704 | 0.9623 |
| **OA** | **0.9642** | **0.9661** | **0.9652** |
