# Peer review of "Inter- and Intra-annual Surface Velocity Variations at the Southern Grounding Line of Amery Ice Shelf from 2014 to 2018"

_The Cryosphere, 2020_

## Referee Comment (RC1) · Chad Greene (Referee) · 24 Jun 2020

**Inter- and Intra-annual Surface Velocity Variations at the Southern Grounding Line of Amery Ice Shelf from 2014 to 2018**

Zhaohui Chi, Andrew G. Klein

Review by Chad A. Greene, Jet Propulsion Laboratory, California Institute of Technology, June 22, 2020.

**General Comments**

This paper presents eleven measurements of ice speed along the southern grounding line of the Amery Ice Shelf, taken in even intervals between early 2014 and early 2019.

I was excited to read the paper because I am deeply interested in the interplay of seasonal and interannual processes at the junction of grounded and floating ice. Given that this paper follows a few years after the author's dissertation on the dynamics of the Lambert Glacier/Amery Ice Shelf system, I looked forward to new insights and rich analysis of the processes that control ice flow here. Unfortunately, this paper offers only a dry list of measurements without any context or discussion of which values might be important, what physical processes might be influencing those values, or what any of them mean.

The Introduction section mentions why ice velocity measurement is important as a general topic, but it's never made clear why I should care about the specific measurements presented in this paper. If the velocity variability tells an interesting or meaningful story, I would hope that the authors would share their expertise in discussing what is causing the variability and what the variability could mean for this particular ice system. I think the paper would be greatly improved if the authors can identify the most important finding of the study, and center the whole paper around that story.

At times, the manuscript presents itself as a methodological paper, but it is not. The techniques described here have been in use for decades, the authors employed standard software to analyze their data, and no technical innovations are presented. If this paper is to be published, I recommend abbreviating the Methods section to include only the information necessary for others to repeat the study, and reworking the overall tone to remove any language that might suggest that this paper presents methodological advancements. By trimming away the unnecessary explanations of the feature tracking technique, it may help readers identify and focus on the main message of the paper.

Setting aside the presentation of the information in this paper, I have some concern that the primary scientific conclusions of this study are potentially the result of finding patterns in measurement noise. Possible sources of error are described in detail in the section below.

The authors have provided enough information that interested readers might be able to follow along with the analysis, but anyone who wishes to perform a follow-on study would undoubtedly prefer the velocity data uploaded as a supplement to the manuscript rather than being left to perform the entire analysis from scratch. Similarly, I'd like the code that was used in this study to be made available, so others may build on this work or understand how the analysis was performed.

Overall, I regret to say I did not find anything insightful in this paper, and I cannot see how the shortcomings could be adequately addressed in revision. No technical advancements are presented; the lack of data or code precludes any follow-on study; there is no analysis or discussion of physical processes that readers might learn from; the reported signal might only be noise; and much lower-noise, higher-quality, longer records of velocity are already freely available and can be used to replicate this kind of study with just a few lines of code.

**Specific Scientific Comments**

**Missing citation:** Given that this paper attempts to put velocity variability of the Amery Ice Shelf into decadal perspective, I'd like to point the authors to work by Matt King, whose PhD thesis and a 2007 paper extended velocity measurements back to 1968.

**Short-term measurements:** It strikes me that winter and summer velocities are measured only as snapshots taken at roughly the same times each winter and each summer. The tacit assumption here is that winter velocities are steady all winter and summer velocities are steady all summer. We know that for many glaciers in Greenland, seasonal velocity variability can happen in fits and spurts. Given that a goal of this paper is to characterize seasonal velocity variability, what is the possibility that these snapshot measurements of velocity are just capturing short-term accelerations, or missing annual events entirely?

In addition to possible aliasing resulting from weeks-long measurements being used to represent full seasons or years, the short timespan of each measurement also introduces high levels of noise. Given that velocity uncertainty stems from displacement uncertainty and there is negligible uncertainty in the measurement of *dt*, an image pair separated by a full year will provide velocity measurements having just 10% of the possible noise associated with the one-month *dt* image pairs used in this study. I recommend using a full year *dt*, at least for the characterization of annual velocities.

**L148-153:** Rock is not the only way to register velocities. There are some large patches of near-zero velocity ice in this region. For each image pair, consider calculating the mean x displacement for all pixels of rock along with all pixels of ice whose mean annual velocity is expected to be less than about 15 m/yr or so. Let's say that gives a mean value of 1.5 m. Now calculate the *expected* mean x displacement of all of those same pixels, let's say that gives a mean of 0.5 m. Such a scenario would suggest there's an average 1 m difference between the

registration of the two images, so subtract that 1 m from dx and then divide by dt to get a more accurate measure of velocity. Do the same in the y direction. This should significantly tighten up the velocity measurements while also providing a good measure of uncertainty which would be the standard deviation of residuals in the near-zero velocity pixels after adjustment.

**L152-153:** The 50 random locations used for uncertainty quantification are highly localized, centered on one rock. While that approach is fair, it relies on an implicit assumption that georegistration errors in Landsat images can be characterized by a single value of offset everywhere within the image. In reality, georegistration error can also take the form of rotation or warping, meaning the tiny patch of 50 random locations may not accurately represent errors elsewhere in the image. Accordingly, if more pixels can be incorporated into the georegistration by tying each image pair to slow-moving ice in addition to rock, this will likely reduce velocity errors throughout each image pair.

The potential for velocity accuracy to vary spatially within an image pair is particularly high when the two images are from different Landsat paths or rows, because the two images are taken from different angles and possibly at different times of day. The manuscript does not explicitly state whether image pairs used in this study are from matching path and row combinations, but the time intervals (non-multiples of 16 day Landsat orbit repeat) listed in Table 1 make it clear that the majority if not all of the image pairs used in this study were from various combinations of paths and rows. This suggests that error estimates from the small rock patch may not represent errors in the rest of the image pairs.

**L169-171:** Annual velocities are taken as the average of winter and summer velocities, which in some could potentially be inadequate, because the measurements only capture about two months of ice motion. If the goal is to characterize annual velocity and separate winter velocities from summer velocities, why not compare summer displacements to the total displacement that occurs throughout the entire year with a ~365 day dt?

**L173:** "Overall, the surface velocities across all three glaciers show a consistent ~ 5% inter-annual increase over the study period (Fig 5)."

I find evidence on the contrary. Figure 5 shows velocity changes that are not monotonic through time, and the only consistent pattern I see is that each velocity profile appears to be offset by a scalar value. This suggests possible contamination by geolocation errors.

Comparing the authors' results to publicly available datasets, **I find no evidence for a consistent 5% increase in annual velocity over the 2014-2018 period**. For example, here is a map of the linear trend in velocity from 2014 to 2018, using annual velocities from the ITS_LIVE dataset (Gardner et al., 2018, Gardner et al., 2019).

[Figure]

Along most of the grounding line I see a slight negative trend on the order of -0.5 m/yr. Multiplying that trend by the full timespan of this study results in a total velocity change of about -0.3% for the Lambert Glacier section of the grounding line.

The map above shows the velocity trend from over 5000 Landsat image pairs acquired between 2014 and 2018. By properly accounting for georeferencing errors in each image pair, then averaging the velocity maps from several hundred image pairs obtained each year, SNR is increased and errors in the annual mosaics of the ITS_LIVE dataset are much lower than errors in any single image pair. In addition, ITS_LIVE includes many long-dt image pairs, which each have much lower uncertainty than the short-dt image pairs described in this manuscript. The much lower noise of ITS_LIVE compared to the measurements presented in this study is apparent when we recreate Figure 5 from the manuscript, using ITS_LIVE data:

[Figure]

The velocity variability presented in Figure 5 of the manuscript is not present in the ITS_LIVE data plotted above, and I suspect that's because the two image pairs per year used in the manuscript contain considerable noise.

I should disclose that I have contributed to the development of the ITS_LIVE dataset, and I am likely biased in its favor. For an unbiased look, we can alternatively look at annual velocities in the publicly available dataset by Mouginot et al. (2017), to which I did not contribute:

[Figure]

Each velocity profile above represents several dozen image pairs in each year, and trends are generally not apparent outside the uncertainty estimates.

The datasets plotted above are all publicly available and extend back as far as 1985. I would also point the authors to the GoLIVE dataset, which provides thousands of image-pair velocities from Landsat 8 that have already been processed and are freely available from the NSIDC (Fahnestock et al., 2015; Scambos et al., 2016).

In case it is helpful, I have attached the MATLAB script I used to create the plots above, using Antarctic Mapping Tools for Matlab (Greene et al., 2017) and the Climate Data Toolbox for Matlab (Greene et al., 2019).

**Sect. 5.3:** I found the distinction between intra-annual and inter-seasonal variability to be unintuitive, difficult to follow, and ultimately not very enlightening. The distinction results in the reporting of several numbers, but it's not clear which of these numbers are important or why. If I have missed something important about these two metrics, please clarify. Otherwise, I suggest sticking to a single metric of subannual variability.

**Technical Corrections**

**L30-32:** States ice flow in Antarctica has been found to be increasingly driven by accelerated surface melting. Is that true? None of the three references at the end of this sentence mention surface melt.

**L37-37:** States the Lambert Basin drains 12.5% of the AIS, citing a map from 1983. Basin outlines, ice thickness, and surface topography are much better constrained today than they were 40 years ago. A quick check using Mouginot's refined IMBIE basin outlines applied to BedMachine ice thickness suggests the Lambert basin actually drains something like 3.5% of Antarctica's grounded ice by area, or 4.2% by volume.

**L43:** Remove "are."

**L61-62:** The sentence beginning, "The extremely frozen environment..." can be removed.

**L111:** The Landsat images used in this paper are in polar stereographic coordinates, and presumably the feature-tracking analysis is performed in these projected "eastings" and "northings." As a result, there's some ambiguity in the phrase "north-south (N/S) and east-west (E/W) components" used on line 111 and throughout the paper. Does it mean geographic coordinates or polar stereographic?

**L114-116:** Confusing and inconsistent mathematical syntax.
- It appears that N/S is a variable name with a mathematical operator in it, making any equation containing N/S difficult to parse.
- The N/S and E/W variables appear later as $NS^2$ and $EW^2$. Be consistent.
- Why not use intuitive variable names for the displacement components, like $dx$ and $dy$?
- Is there a reason to use the acronym $SQRT$ in the equation instead of standard mathematical notation?
- The $t$ variable is clearly defined in words on line 117, but it may be more intuitive for readers to see it as $dt$, or $\Delta t$, or $t_2$-$t_1$ as it represents a difference in time.

**L119:** The sentence "Due to providing sharp surface texture..." needs attention for syntax.

**L141-143:** The passive voice is somewhat ambiguous in the phrase "uncertainty of imagery co-registration has been minimized." Does this mean you have performed some minimization procedure? If so, describe it and provide a rationale. Or does it mean you are relying solely on the georegistration performed in the creation of the L1GT product?

**L143:** Period at the end of the sentence.

**L144-146:** There are some syntax errors in the sentence beginning "Hence,..." but I think the sentence can be deleted entirely without detriment to the manuscript.

**L182-184:** If I am reading this sentence correctly, intra-annual variability is defined as summer minus winter velocity over summer velocity, or $(v_s-v_w)/v_s$, but the following sentence says negative intra-annual variation indicates that winter velocity is less than summer velocity. I think it's the other way around?

**Subsection 5.5** is about measurement accuracy, and would feel more at home under the Methods section than the Results section. I suggest this because it's a characterization of the method that was used, not a final result of the study.

**L240:** "The overall accuracy of feature tracking is approximately 96.5%" I understand the concept of SNR, but I do not understand how accuracy can be defined as a percentage in this context or what it means to the interpretation of the results. Please clarify.

**L255:** Remove "relevant" or give context for what is meant by this word.

**L271-272:** "No obvious velocity variations appear to exist between summers and winters from our multiyear average measurements." Perhaps I am misunderstanding this sentence, because it appears to contradict all previous statements that seasonal variability does exist here.

**L279:** "ground line" should be "grounding line".

**L287-289:** The sentence beginning "While limited..." has some missing words. I think it should contain something like "...or feature tracking are available." However, fixing the syntax will not completely fix this sentence. It currently makes a vague statement that implies that no data or very little data are available before the 2014 start of this study; yet, velocities by Mouginot et al. are publicly available and extend back to the mid 2000s and ITS_LIVE velocities are available here as early as the mid 1980s. Going back even further, King et al. (2007) reported velocity changes of the Amery Ice Shelf back to the 1960s. If there is a critical gap in the historical data record, be clear about exactly what that gap is, state what we *don't* know, and help us understand what we *do* know about past velocity variability.

**L299-301:** This paragraph doesn't contain any conclusions that are unique to this study, and it can be removed without detriment to the manuscript.

**Figure 2** is not necessary, as the mechanics of feature tracking are not integral to understanding the dynamic variability reported in this study, and more intuitive versions of this figure have been published before, such as Fig. 2 of Scambos et al., 1992. If the intent of this figure is to teach readers how feature tracking works, then I recommend using words in the caption to help explain the figure. Please also convert this to a higher resolution image and clarify whether the two largest offset boxes are offset in space, or if the offset is intended to indicate time in the third dimension.

In addition, if this figure is included in the final publication, it should be properly attributed to its original author. The caption in this manuscript attributes it to Chi 2012, yet in Chi 2012 the figure is attributed to Huang and Li, 2009.

**Figures 4a, 5, 6, and 9:** It would be helpful to see the measurement uncertainty estimates on the velocity profiles, perhaps as shaded regions. That way viewers can directly see how to interpret the differences in velocity from year to year, in the context of estimated measurement error.

**Figure 9:** Two velocity profiles are shown in panel A: the feature-tracking velocities attributed to the years 1999-2016, and InSAR velocities attributed to 1996-2016. What's unclear is how these velocities were obtained over two decades. Do they represent some average velocities? If so, how many velocity profiles were calculated over these decades, how were they combined, and assuming data were not equally available throughout these decades, what's the nominal date of the measurements? Also, the figure caption should read MEaSUREs, not MEaSURE.

**Table 1:** What are the path/row combinations of each image?

**References**

Fahnestock, M., T. Scambos, T. Moon, A. Gardner, T. Haran, and M. Klinger. 2015. Rapid large-area mapping of ice flow using Landsat 8, Remote Sensing of Environment. 185. 84-94. https://doi.org/10.1016/j.rse.2015.11.023

Gardner, A. S., M. A. Fahnestock, and T. A. Scambos, 2019: ITS_LIVE Regional Glacier and Ice Sheet Surface Velocities. Data archived at National Snow and Ice Data Center; doi:10.5067/6II6VW8LLWJ7.

Gardner, A. S., G. Moholdt, T. Scambos, M. Fahnstock, S. Ligtenberg, M. van den Broeke, and J. Nilsson, 2018: Increased West Antarctic and unchanged East Antarctic ice discharge over the last 7 years, Cryosphere, 12(2): 521–547, doi:10.5194/tc-12-521-2018.

Greene, Chad A., David E. Gwyther, and Donald D. Blankenship. "Antarctic Mapping Tools for MATLAB." Computers & Geosciences 104 (2017): 151-157.

Greene, Chad A., et al. "The Climate Data Toolbox for MATLAB." Geochemistry, Geophysics, Geosystems 20.7 (2019): 3774-3781.

King, Matt A., et al. "Velocity change of the Amery Ice Shelf, East Antarctica, during the period 1968–1999." Journal of Geophysical Research: Earth Surface 112.F1 (2007).

Mouginot, J., E. Rignot, B. Scheuchl, and R. Millan. 2017. Comprehensive Annual Ice Sheet Velocity Mapping Using Landsat-8, Sentinel-1, and RADARSAT-2 Data, Remote Sensing. 9. Art. #364. https://doi.org/10.3390/rs9040364

Scambos, Theodore A., et al. "Application of image cross-correlation to the measurement of glacier velocity using satellite image data." Remote sensing of environment 42.3 (1992): 177-186.

Scambos, T., M. Fahnestock, T. Moon, A. Gardner, and M. Klinger. 2016. Global Land Ice Velocity Extraction from Landsat 8 (GoLIVE), Version 1. Boulder, Colorado USA. NSIDC: National Snow and Ice Data Center. doi: https://doi.org/10.7265/N5ZP442B.

**MATLAB Code**

To create the figures in this review.

```
% This script recreates the interannual velocity variability
% analysis in Chi & Klein, 2020 https://doi.org/10.5194/tc-2020-99
%
% Written by Chad A. Greene, June 2020.
% Uses Antarctic Mapping Tools for Matlab and Climate Data Tools for Matlab,
% with MOA grounding line dataset, MEaSUREs annual velocities, and ITS_LIVE
% annual velocities.

%% Load data

% Some coordinates bounding the region of interest:
xi =[1667314.32 1727419.46];
yi =[646502.61 756695.37];

% Load the 2009 MOA grounding line data and trim to region of interest:
load 'moagl2009.mat'
[glx,gly] = ll2ps(gllat,gllon);
in = inpsquad(glx,gly,xi,yi);
glx = glx(in)';
gly = gly(in)';

% Distance along grounding line:
d = pathdistps(glx,gly,'km');

% Get ITS_LIVE velocity data along grounding line:
yrs = 1985:2018; % start at 2014 to match Chi & Klein, or 1985 for full
record.
v = itslive_interp('v',glx,gly,'years',yrs);
v_err = itslive_interp('v_err',glx,gly,'years',yrs);

%% Recreate Figure 5 with ITS_LIVE data:

% Define colormap:
cm = parula(length(yrs)); % for a long time record
%cm = rgb({'dark blue';'cyan';'green';'orange';'dark red'}); % to match the
manuscript

figure
hold on

% Plot the velocity profile for each year of data:
for k= 1:length(yrs)
   try % any year with all nans will throw an error, so "try" is an easy to
keep things moving.
      [hl(k),hp(k)] =
boundedline(d,v(:,k)/365.25,v_err(:,k)/365.25,'nan','gap','alpha','color',cm(
k,:));
   end
end

% Move the uncertainty estimate patches to the bottom of the stack for
clarity:
```

```matlab
for k = length(yrs):-1:1
    try
        uistack(hp(k),'bottom')
    end
end
axis tight
box off
colorbar
caxis([yrs(1)-0.5 yrs(end)+0.5])
ylabel 'velocity m/day'
xlabel 'distance along gl (km)'
colormap(cm)
axis([20 100 0 3])

%export_fig amery_velocity_profile_itslive_1985-2018.png -r300

%% Recreate Figure 5 with Mouginot data

% Load the data:
[vm,tm] = measuresann_interp('speed',glx,gly);
[vx_err,vy_err,~] = measuresann_interp('error',glx,gly);
vm_err = hypot(vx_err,vy_err);
vm = squeeze(vm);
vm_err = squeeze(vm_err);

% Trim to 2014-1026:
ind = tm>=2014;
tm = tm(ind);
vm = vm(:,ind);
vm_err = vm_err(:,ind);

% Define the colormap to match the manuscript:
cm = rgb({'dark blue';'cyan';'green';'orange';'dark red'});

figure
hold on

for k= 1:length(tm)
    try
        [hl(k),hp(k)] =
boundedline(d,vm(:,k)/365.25,vm_err(:,k)/365.25,'nan','gap','alpha','color',c
m(k,:));
    end
end
for k = length(tm):-1:1
    try
        uistack(hp(k),'bottom')
    end
end
axis tight
box off
cb = colorbar;
caxis([tm(1)-0.5 tm(end)+0.5])
set(cb,'ytick',tm)
ylabel 'velocity m/day'
xlabel 'distance along gl (km)'
colormap(cm(1:length(tm),:))
```

```matlab
axis([20 100 0 3])

% export_fig amery_velocity_profile_mouginot.png -r300

%% Make a trend map with ITS_LIVE data

% Load data
yrs = 2014:2018;
[v,x,y] = itslive_data('v',glx,gly,'buffer',25,'years',yrs); % velocity
w = 1./itslive_data('v_err',glx,gly,'buffer',25,'years',yrs).^2;% Weights

% Reashape to 2D:
v2 = cube2rect(v);
w2 = cube2rect(w);

% Preallocate 2D velocity trend matrix:
v_tr2 = nan(1,size(v2,2));

% Loop through each grid cell:
for k = 1:length(v_tr2)

   % Compute the weighted dv/dt for this grid cell:
   tmp = polyfitw(yrs',v2(:,k),1,w2(:,k));

   % Record it:
   v_tr2(k) = tmp(1);
end

% Reshape the trend:
v_tr = rect2cube(v_tr2,size(v));

% Get surface elevation for context:
[X,Y] = meshgrid(x,y);
z = bedmachine_interp('surface',X,Y);

figure
h = surf(x,y,z,v_tr);
shading interp
view(2)
axis tight off
daspect([1 1 1])
caxis([-10 10])
shadem(2) % hillshade
h.ZData = h.ZData-max(h.ZData(:)); % push down
hold on
ax = axis;
q = itslive_quiver('color',rgb('green'));
axis(ax);
plot(glx,gly,'k','linewidth',2)
cmocean bal
cb = colorbar;
ylabel(cb,'velocity trend 2014-2018 (m/yr)')

%export_fig amery_velocity_trend_2014-2018.png -r300

%% Velocity trend profile and count
```

```
% Note: to get total image pair counts, try:
count =  itslive_data('count',glx,gly,'buffer',25,'years',yrs);
figure
imagescn(x,y,sum(count,3))
hold on
plot(glx,gly,'k','linewidth',2)
colorbar

figure
subplot(2,1,1)
plot(d,interp2(x,y,v_tr,glx,gly))
axis tight
box off
ylabel 'velocity trend 2014-2018 (m/yr)'
xlim([20 100])
hline(0,'color',rgb('gray'))

subplot(2,1,2)
plot(d,interp2(x,y,sum(count,3),glx,gly));
axis tight
box off
ylabel 'image pair count'
xlim([20 100])
xlabel 'distance along GL (km)'
```

---

## Referee Comment (RC2) · Anonymous Referee #2 · 4 Sep 2020

**Review of:**

Inter-and Intra-annual Surface Velocity Variations at the Southern Grounding Line of Amery Ice Shelf from 2014 to 2018

*Zhaohui Chi and Andrew G. Klein*

**General comments**

I find the manuscript not suitable for publication in the present form. The two major shortcomings in my opinion concern: a) the reliability of the measurements and b) the lack of clear conclusions and of result interpretation.

The measurements are not very robust, since they rely on very few (11) Landsat-8 image pairs, and on some processing choices, which could further increase the sensitivity to measurement errors (e.g. short temporal separations and lack of spatial averaging). In particular the effects of measurement biases could be significant and only partially captured by the quality assessment approach.

The conclusions of the paper are unclear. In several parts of the paper the authors state with confidence that significant velocity variations are observed in their data, but this seems to be contradicted or at least significantly softened by the conclusions section. Furthermore, interpretation of the results is simply missing, and no hypotheses are put forward to explain the observations.

My detailed comments are provided in the following.

**Specific comments**

**L86-93:** You select a single image pair within each season, and state at L90 that the preferred temporal interval for each image pair is one month. This raises several questions, which would deserve further consideration and discussion:

- Can the average velocity over a month be assumed to represent the average velocity of a whole season in this region?
- What are your criteria for data selection, i.e. why do you select exactly those pairs? Why not process all the available cloud-free pairs?
- Why not increase the time interval between images? Based on equation (1), the larger the time interval the less sensitive you will be to errors in the measured N/S and E/W displacements.

**L95-97:** Can the Grounding Line Location be assumed to be unchanged between 2009 and the 2014-2018 period? Could an imprecise location have any impact on your results?

**L102-117:** I find this description redundant, as well as Fig. 2. The method is well established, and it would be sufficient to reference the relevant papers, which are already included in the reference list.

**L126-127:** If I understood correctly you carried out feature tracking only on "a set of location reference points" along the AIS grounding line. Would it reduce measurement noise to carry out the measurements on a 2D grid to allow for some spatial averaging of the measurements?

**L148-153:** This kind of error estimate will provide a lower bound for the actual errors, since the features which are cross-correlated on stable ice are not necessarily representative of those which are causing the cross-correlation peaks on faster flowing ice. A better error characterization would be provided by the standard deviation of measurements carried out using different image pairs.

**L173-174:** You state that Fig. 5 shows a "consistent ~5% inter-annual increase over the study period". To me this is not at all obvious. Based on both Fig. 5 and Table 2 there seems to be a decrease between 2015 and 2016 and the increases observed between the other years are quite variable in magnitude.

**L251-253:** You mention that the "contribution of the measured magnitudes and directions of these uncertainties in our computed velocity and variability assessments did not lead to any significant adjustments to our measurements". However, to me it seems like the error biases you show in Fig. 8, left panel, which vary between 0.1 and 0.25 m/d, are of the same order of magnitude of your inter-annual variabilities in Table 2, and would therefore be significant.

**L255-256+L267:** to me it seems a contradiction that you first state that there is little previous research, and then you state that your results are consistent with previous studies. At L255-263 you don't refer to other studies in the time period you consider, namely 2014-2018, so which studies do you refer to at L267 when you state "consistent with previous studies"?

**L310-311:** You state that "over the five year study period general stability in surface velocities were observed". This seems to contradict several other statements, such as L266 and L173, as well as the abstract itself, where you state that there is a variability.

**Technical corrections**

**Table 1:** For someone to replicate your study, it would be useful for you to add the paths of each acquisition.

**L179:** In Table 2 you state 0.29 m/d, whereas here you state 0.31 m/y. These values should agree.

**L183-184:** Negative intra-annual velocity would mean winter velocity > summer velocity based on the definition you give here.

**L240-241:** It seems you are defining accuracy as the SNR expressed as a percentage. This is misleading and I would rather just refer to the SNR values, since you have defined the SNR in section 4.

**L269:** Both "m per year" and "m/yr" are used. Use the same notation throughout the paper.

**English language:** In all section besides the Introduction, the English language needs some minor revisions.